# The Effectiveness of a Single Dry Needling Session on Gait and Quality of Life in Multiple Sclerosis: A Double-Blind Randomized Sham-Controlled Pilot Trial

**DOI:** 10.3390/healthcare12010010

**Published:** 2023-12-19

**Authors:** Alberto Javier-Ormazábal, Montserrat González-Platas, Alejandro Jiménez-Sosa, Pablo Herrero, Diego Lapuente-Hernández

**Affiliations:** 1Division of Physiotherapy, Hospital Universitario de Canarias, Carretera Ofra S/N, 38320 San Cristóbal de La Laguna, Santa Cruz de Tenerife, Spain; albert.jaor@gmail.com; 2Research Institute of Biomedical and Health Sciences, Universidad de Las Palmas de Gran Canaria, C. Juan de Quesada 30, 35001 Las Palmas de Gran Canaria, Las Palmas, Spain; montserrat.gonzalezplatas@gmail.com; 3Division of Neurology, Hospital Universitario de Canarias, Carretera Ofra S/N, 38320 San Cristóbal de La Laguna, Santa Cruz de Tenerife, Spain; 4Research Unit, Hospital Universitario de Canarias, Carretera Ofra S/N, 38320 San Cristóbal de La Laguna, Santa Cruz de Tenerife, Spain; ajimenezsosa@gmail.com; 5Department of Physiatry and Nursing, Faculty of Health Sciences, University of Zaragoza, C/Domingo Miral s/n, 50009 Zaragoza, Zaragoza, Spain; d.lapuente@unizar.es; 6iHealthy Research Group, IIS Aragon, Avda San Juan Bosco 13, 50009 Zaragoza, Zaragoza, Spain

**Keywords:** multiple sclerosis, dry needling, gait performance, walking capacity

## Abstract

Introduction: Gait disorders are a major cause of disability and reduced health-related quality of life in people with multiple sclerosis (pwMS). Dry needling (DN) has demonstrated positive results to improve gait parameters in patients with stroke. The main aim of this study was to evaluate the effect of a single session of DN in the gait performance of pwMS. Methods: A double-blind parallel randomized sham-controlled pilot trial was conducted. Study participants received a single session of active DN or sham DN in the gastrocnemius medialis muscle. Pre-treatment and immediately post-treatment measurements were taken, as well as at one and four weeks after the intervention. Outcomes related to gait performance (Timed 25-Foot Walk), self-perceived walking capacity (Multiple Sclerosis Walking Scale), risk of falls (Timed Up and Go test), disability level (Expanded Disability Status Score) and quality of life (Multiple Sclerosis Quality of Life-54 questionnaire and Analogic Quality of Life scale) were evaluated. Results: 18 patients who had multiple sclerosis participated in the study. The group who received active DN showed within-group significant statistical differences immediately after treatment for gait performance (*p* = 0.008) and risk of falls (*p* = 0.008), as well as for self-perceived walking capacity at one week (*p* = 0.017) and four weeks (*p* = 0.011) and quality of life at four weeks (*p* = 0.014). Regarding the comparison between groups, only significant results were obtained in the physical domain of the quality of life at four weeks (*p* = 0.014). Conclusions: DN seems to be a promising therapeutic tool for the treatment of gait disorders in pwMS. However, when results were compared with sham DN, no differences were found.

## 1. Introduction

Multiple Sclerosis (MS) is the most common inflammatory neurological disease in young adults [1]. This condition can include a great variety of neurological manifestations such as optic neuritis, diplopia, sensory loss, spasticity, limb weakness, gait ataxia, loss of bladder control and cognitive dysfunction, amongst others [2]. Its etiology is complex and multifactorial, with genetic and environmental interactions predominant in women (3:1) and considered the second cause of disability in young adults (25–30 years old) [3]. The total estimated prevalence rate of MS for the past three decades in Europe is 83 cases per 100,000, with a mean annual incidence rate of 4.3 cases per 100,000 [4]. MS puts forward a very significant challenge to both healthcare systems and global society due to productivity losses and caregiver burdens, with the costs of drugs considered one of the main associated costs [5]. In Spain, the total cost associated with MS is estimated at EUR 1395 million per year, and the mean annual cost per patient of EUR 30,050, as well as a great impact on the quality of life of patients, with an annual loss of up to 13,000 quality-adjusted life years [6].

MS may affect the brain, brainstem, spinal cord and optic nerves and may lead to different balance and gait impairments, as well as increasing the risk of falls [7]. Unlike other neurological diseases, the progression of disability in pwMS can vary from day to day and throughout the day due to changes in temperature, fatigue and activity. Therefore, walking impairments may change gradually over time or may appear with a sudden onset, with about 80% of pwMS having walking difficulties during their illness. Gait disorders are a hallmark of MS [8], with a decreased cadence and speed and an increased stride-to-stride variability [9,10] being the most relevant. Therefore, the improvement in walking ability is a primary goal for rehabilitation treatment in pwMS [11]. Ambulation difficulty has functional and emotional consequences in pwMS and is a major cause of disability and reduced health-related quality of life. As symptoms worsen, quality of life worsens and healthcare utilization increases [12,13]. 

The main focus of rehabilitation treatments in pwMS is improving weakness, balance and gait [14,15]. Dry Needling (DN) has been demonstrated to be effective for improving gait in patients with stroke [16,17], whereas, in other populations such as Parkinson’s Disease [18] or Spinal Cord Injury [19], its effect is still unclear. To date, only two case reports have been carried out to study the effect of DN on the gait of pwMS, suggesting that it may be an intervention that improves mobility and gait speed [20,21]. However, there is no existence of any randomized controlled trial evaluating the effect of DN on the gait of pwMS. Therefore, the main aim of this clinical trial was to evaluate the effect of a single session of DN in their gait performance. Secondarily, it was also to assess the effects of DN on self-perceived walking capacity, the risk of falls, the level of disability and the quality of life.

## 2. Materials and Methods

### 2.1. Study Design

A double-blind parallel randomized sham-controlled pilot trial was conducted. This study was designed according to Consolidated Standards of Reporting Trials guidelines (CONSORT) and all the procedures were applied in accordance with the Declaration of Helsinki. The ethical approval was obtained from the Clinical Research Ethics Committee of Hospital Universitario de Canarias, Canary Islands, Spain (CHUC_2023_40). The study was prospectively registered in ClinicalTrials.gov (NCT05956119). All patients signed an informed consent document before participating. 

### 2.2. Participants

The study was carried out in the neurology service of Hospital Universitario de Canarias between July and October 2023. The inclusion criteria were the following: (1) diagnosed with multiple sclerosis in a period of more than two years prior to the start of study recruitment, (2) aged between 18 and 60 years, and (3) having a score > 2 on the Expanded Disability Status Score and specifically >2 in the “Pyramidal” domain, with the gastrocnemius medialis affected. The exclusion criteria were the following: (1) an insurmountable fear of needles, (2) presenting a specific outbreak of the disease in the last two months before starting the study, and (3) receiving other concomitant treatments or changes in the usual treatment which may interfere with the study results. 

### 2.3. Randomization and Blinding 

Participants were informed of the study and agreed to participate by signing the informed consent form. Following the baseline assessment, participants were randomly assigned to each group with a 1:1 allocation ratio. Randomization was achieved using an online research randomizer sequence generator (http://www.randomizer.org (accessed on 9 May 2023)) by an independent researcher. After randomization, the physical therapist performed the treatment according to the assignments. Assessment and data analysis were carried out by a researcher who was blinded to participant’s treatment allocation. Participants were also blinded and were asked not to reveal any information related to the treatment to the researcher performing the assessments. 

### 2.4. Interventions

The intervention group (IG) received one session of DN in the gastrocnemius medialis muscle (see Figure 1), whereas the control group (CG) received one session of sham DN in the same muscle. Participants didn’t receive any other physiotherapy treatments during the study. Both interventions were performed by the same physical therapist with five years of experience with DN. Before performing the active DN or the sham intervention, the skin was disinfected with a gauze and alcohol. The gastrocnemius medialis was placed in submaximal stretch position, following the criteria for DN in neurological patients [22]. The treatment consisted of a single session of DN with disposable stainless-steel needles (0.3 mm × 40 mm, Agupunt, Barcelona, Spain) with a pistoning technique performed at 1 Hz for approximately 30 s for the IG, whereas for the CG a single session of sham DN using a monofilament, without introducing it through the skin but simulating the sensation of the needle for 30 s, was carried out. To offer the greatest possible blinding to the study, all patients were placed face down without visualizing the technique being performed and the procedure was replicated for both groups, mimicking the context when performing any DN treatment (e.g., hemostatic compressions after procedures were identical in both interventions). Patients were treated only on one side, the one exhibiting more spasticity. 

### 2.5. Outcomes

Outcome measures were assessed before treatment, immediately after treatment and at one week and at four weeks follow-up by a neurologist specialized in MS who was blinded to the group allocation. The outcomes that were measured at each of the follow-up moments are detailed in Figure 2. 

### 2.6. Primary Outcome

Gait performance. The Timed 25-Foot Walk (T25FW) was used to measure gait performance. This is considered the “gold standard” to assess gait speed in pwMS. The test measures the time in seconds that the patient needs to walk 25 feet (7.5 m). The test was conducted twice, and the mean value was calculated to improve its validity and reliability [23]. The T25FW is considered reliable to assess ambulation changes in pwMS [24]. 

### 2.7. Secondary Outcomes

Self-perceived walking capacity. The Multiple Sclerosis Walking Scale (MSWS-12) was used to quantify the impact of MS on the individual’s walking ability [25]. Each item is scored from 1 (no limitation) to 5 (extreme limitation), with a total score ranging from 12 to 60, where higher values show higher impairment. MSWS-12 shows properties which make it suitable for use in clinical practice, specifically for patients with medium to high levels of walking disability [26]. Test–retest reliability has been confirmed, and MSWS-12 is considered a valid, reliable and responsive self-reported measurement of walking impairment in pwMS [27].

Risk of falls. The Timed up and go test (TUG) was used to assess participants’ potential risk of falls. This test was designed initially to measure the probability of falls among older adults but has been validated also for other populations such as pwMS [28]. Participants sit in a chair with their backs supported by the back of the chair and their arms resting on the armrests. Participants are asked to stand up and walk 3 m, to turn on themselves (360°) and walk back to the chair and sit down again. The participants performed the test twice and the mean value was chosen. The TUG has shown to be reliable for assessing the walking capacity and the general mobility of pwMS with mild disability [28]. Test–retest reliability and reproducibility have been also confirmed.

Disability level. The level of disability in the different aspects of the person was measured by the Expanded Disability Status Score (EDSS). This scale is used to measure eight different domains from 0 to 10: (1) visual (eye fundus, scotoma, diopters, etc.); (2) brainstem (nystagmus, dysphagia, dysarthria, etc.); (3) pyramidal (extremity strength, reflexes, etc.); (4) cerebellum (ataxias, dysmetria); (5) sensitive (cutaneous, vibratory, deep sensitivity); (6) vesical/intestinal (micturition dysfunction, intestinal, urgency, incontinence, catheterization, etc.); (7) mental and (8) ambulation. EDSS is a standard instrument for assessing pwMS and has been used in clinical trials to assess the effectiveness of clinical interventions in pwMS [29]. The test was carried out by a neurologist specializing in MS. 

Quality of life. To evaluate the degree of quality of life of the person, two scales were used. On the one hand, it was measured by The Multiple Sclerosis Quality of life-54 questionnaire (MSQol 54). This is a specific quality of life questionnaire for MS. This is a self-assessment questionnaire composed of 54 items, divided into a physical and a mental domain. The range goes from 0 to 100, where 100 is the highest degree of quality of life. The Spanish version of the MSQoL 54 instrument has shown to be a valid and reliable instrument for measuring quality of life in pwMS [30,31]. On the other hand, it was measured by the Analogic Quality of Life scale (AQL). In this test, the patient indicates his or her quality of life on a line or scale, ranging from the ‘best possible quality of life’ to the ‘worst possible quality of life’ [32]. Earlier studies have demonstrated this to be a valid and reliable measure of quality of life [33]. It also has a good validity and an excellent reliability and because of this it is used to measure the global quality of life in clinical trials [34]. Both scales were carried out by a blinded evaluator. 

### 2.8. Data Analysis

Statistical analysis was performed using SPSS statistics version 25.0 (SPSS Inc., Chicago, IL, USA). Descriptive results for continuous data were expressed as means and standard deviations, while nominal data were described as percentages. Comparisons between groups at baseline were performed with the Mann–Whitney U test. Global within-group comparisons were performed using the Friedman test. If the overall test was significant, comparisons were carried out with the baseline values using the Wilcoxon test. Differences between groups in the treatment effect (comparison pre- and post-intervention) and in the different follow-ups (after 1 and 4 weeks) were analyzed with the Mann–Whitney U test. In all cases, *p* values <0.05 were considered statistically significant.

## 3. Results

18 patients (45.71 ± 7.64 years; women 62%) who met the eligibility criteria agreed to participate and were randomized into the DN group (*n* = 9) or the sham DN group (*n* = 9). The reasons for ineligibility of the subjects are specified in the study flowchart (Figure 3). There were no significant differences between groups regarding demographic and clinical characteristics at baseline (*p* > 0.05) (Table 1). All participants completed the treatment intervention. During the trial, no adverse events were registered. 

Gait performance, measured with the T25FW test, showed within-group significant statistical differences for the active DN group only immediately after treatment (*p* = 0.008), although this effect was not maintained after one week (*p* = 0.512) and four weeks (*p* = 0.260). The risk of falls, evaluated with the TUG test, also decreased immediately after treatment (*p* = 0.008), but the effects were not maintained after one week (*p* = 0.110) and four weeks (*p* = 0.678). On the other hand, self-perceived walking capacity, measured with the MSWS-12, had significant within-group differences at one-week (*p* = 0.017) and at four-week (*p* = 0.011) follow-ups, as well as the quality of life measured with the physical domain of the MSQol-54 at four weeks (*p* = 0.014). However, the disability level assessed with the EDSS did not have significant changes at any time points (see Table 2).

When both groups were compared, there were only significant improvements at the four-week follow-up for the physical domain of the MSQOL-54 in favor of the intervention group (*p* = 0.014). The results for all the measures and the time points are summarized in Table 2.

## 4. Discussion

To our knowledge, this is the first randomized and controlled study that aims to evaluate the effect of a single session of DN on the gait performance and walking capacity of pwMS. In this sense, we have only found a protocol of a randomized clinical trial that has yet to have its results published [35]. Our results show that DN in pwMS has positive effects in gait performance and in the risk of falls immediately after the application of a single session, as well as in self-perceived walking capacity at one and four weeks. However, when results were compared with sham DN, no differences were found. Therefore, although the study results may suggest a positive effect of DN, the risk of underpowering to detect a difference between groups cannot be ruled out. 

There is much evidence that supports the effectiveness of DN in the treatment of spasticity in neurological disorders [36,37]; however, its use in treatments aimed at improving gait has hardly been studied, especially in pwMS. A previous case report obtained results similar to ours, confirming our initial hypothesis that DN seems to improve walking speed measured with the T25FW [18]. However, in our study, the minimally important clinical difference in T25FW for pwMS, established as changes from baseline of around 17.2% to 20% or 2.7 s [38,39,40], has not been achieved.

Although the impact that DN has on gait in different populations of patients with neurological impairments needs to be further researched, it is known that it decreases the abnormal electrical activity characterizing MTrPs in spastic muscles [41] and provokes a mechanical disruption of the dysfunctional endplate area [42]. In patients with stroke, DN has shown to have effects on gait speed [17], whereas in other neurological conditions such as pwMS, it is yet unknown if DN may have clinical effects. When assessing pwMS, it is important to consider also qualitative changes in gait, as DN has shown also to have neuromodulation effects at different levels [43,44]. This involves that some patients may experience improvements in self-perceived walking capacity despite not achieving relevant clinical changes in the quantitative parameters of gait. 

Because of the effects that DN may have, other aspects evaluated directly related to walking and functionality were self-perceived walking capacity, risk of falls and disability level, all of which are involved in the progression and prognosis of the disease and the person who suffers from it. The incidence of falls in pwMS is estimated to be more than 50% at least once within a 3 month period, like in adults older than 80 years, with the physical, social and psychological impact that this entails for the individual [45]. Amongst the factors associated with the risk of falls or predictors of future falls, altered balance, reduced walking speed, a higher level of disability, decreased self-efficacy and perceived balance confidence stand out [46,47,48,49]. In this sense, the results shown in our study seem encouraging, despite not having found differences between groups.

Increased fatigue and low walking activity levels are significantly associated with increased fall risk and lower quality of life in people with MS, as well as with increased socioeconomic costs [50,51]. Regarding quality of life, in our study, this was the only outcome in which differences were observed when comparing active DN and sham DN, with improvements occurring in favor of the intervention group at four weeks in the physical domain, measured with the MSQoL-54 scale. Similar improvements have also been found after the application of DN in patients with subacute [52] and chronic stroke [22]. In pwMS, these results in quality of life have also been found in the study by Del Pilar Pérez-Trujillo et al. [21], in which 12 sessions of DN significantly improved quality of life.

The exact number of sessions of DN necessary to have clinical benefits in different outcomes has not yet been defined, with a wide variety of protocols having been published. Therefore, this type of study is important since it demonstrates the potential effectiveness of a single DN session in different clinical outcomes and can establish the future protocols in which the exact number and time of treatment required for the treatment of musculoskeletal disorders like MS can be determined. Similar results have been found in the application of DN in patients with stroke [16], although the participants in these studies were treated in more muscles and received more sessions. In pwMS, the only notable study is the one previously mentioned by Del pilar Pérez-Trujillo et al. [21] in which the technique was performed on more muscles of the lower limbs and not only on the gastrocnemius medialis muscle. 

The main strength of the study is that this is the first randomized, double-blind pilot study that analyzes the effect of DN in pwMS, which also includes a four-week follow-up. However, there are a few limitations that must be considered when interpreting the results. Firstly, the greatest limitation is the reduced sample size, which may explain not finding differences between groups despite having a positive effect in the active DN group. pwMS present a great variability in symptomatology, degree of disability and disease progression, which makes a large sample size necessary to avoid these biases. Nevertheless, other similar studies in neurological populations also present limited samples [52,53,54,55,56,57] and this pilot study builds the framework for future studies focused on the management of gait disorders in pwMS, allowing us to use the values obtained for future sample size calculation. On the other hand, another aspect that may have affected the study population is that the experimental part of the study was carried out in summer, with extremely high temperatures, since it has been observed that this can significantly affect physical and mental abilities of pwMS [58,59]. Finally, other possible explanations for not finding differences between groups would be that only a single DN session was performed and only a single muscle was needled, the gastrocnemius medialis in this case.

Future studies that address the same topic should include larger and more homogeneous samples, evaluating the effects of needling more muscles and more sessions, also with longer follow-up times to evaluate the effect of DN also in the long term. Moreover, other invasive techniques should be evaluated, as it is the case of percutaneous neuromodulation, which showed promising effects in the treatment of musculoskeletal disorders in pwMS [60], although cause–effect relationships cannot be established due to the fact that it is a case report. Furthermore, it would be interesting to carry out cost-effectiveness studies of DN in pwMS in comparison with other therapies or even with drugs such as fampridine in the same way that has been done in chronic stroke [61], as well as the value of adding DN to a program of neurorehabilitation, something already studied in the population with stroke [53].

## 5. Conclusions

DN seems to be a promising treatment to improve gait performance, risk of falls and self-perceived walking capacity. However, a single session of DN performed in the gastrocnemius medialis does not seem to have enough effect to achieve significant changes when compared to sham DN. On the other hand, a single session achieved positive changes in the physical domain of quality of life four weeks after its application. 

## Figures and Tables

**Figure 1 healthcare-12-00010-f001:**
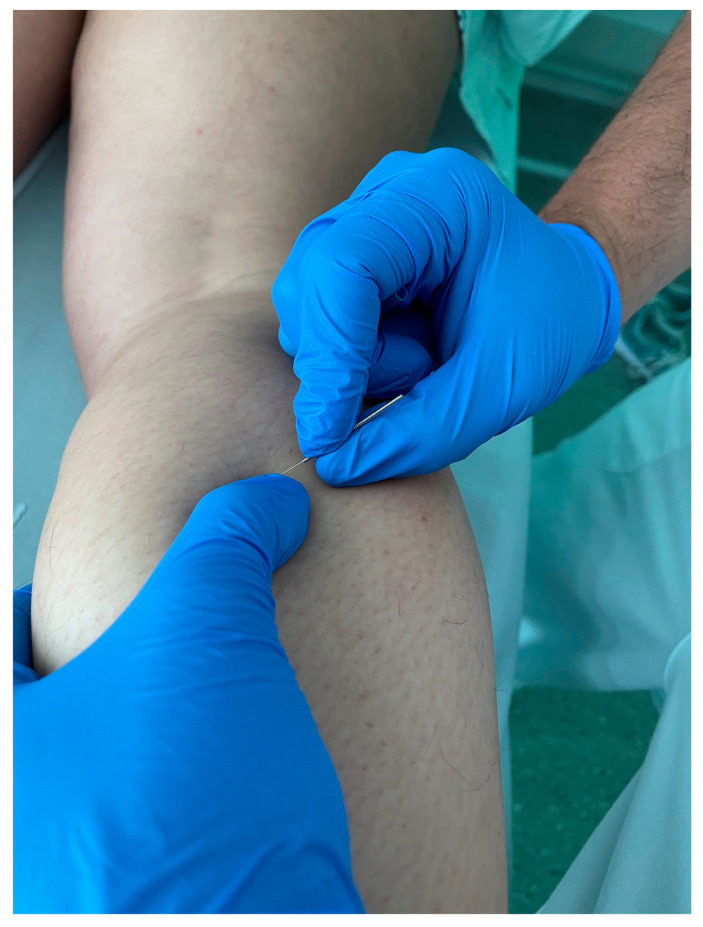
Dry needling on gastrocnemius medialis muscle.

**Figure 2 healthcare-12-00010-f002:**
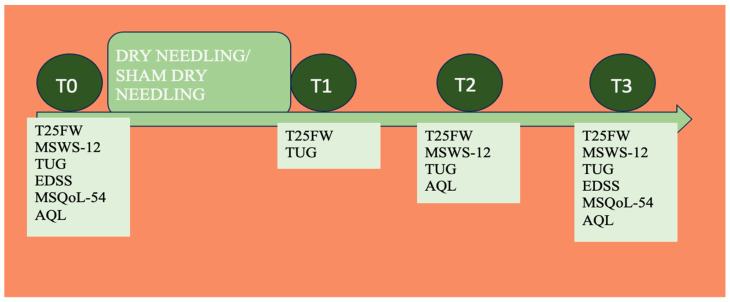
Different time points (T0: just before intervention; T1: just after intervention; T2: 1 week after intervention; T3: 4 weeks after intervention) and outcomes measured in each of them. AQL: Analogic Quality of Life scale; EDSS: Expanded Disability Status Score; MSQol 54: The Multiple Sclerosis Quality of life-54 questionnaire; MSWS-12: Multiple Sclerosis Walking Scale; T25FW: Timed 25-Foot Walk; TUG: Timed up and go test.

**Figure 3 healthcare-12-00010-f003:**
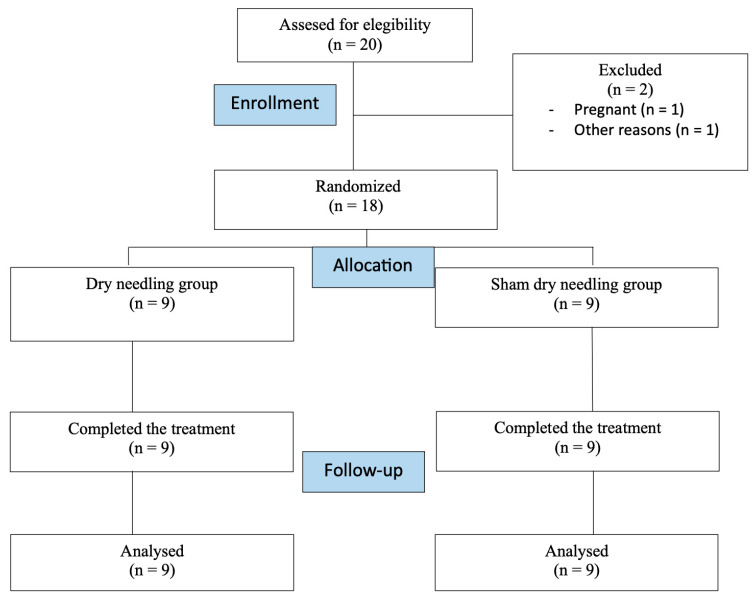
CONSORT 2010 flow diagram of participants throughout the course of the study.

**Table 1 healthcare-12-00010-t001:** Baseline clinical and demographic data of the participants.

	IG (Active Dry Needling, *n* = 9)	CG (Sham Dry Needling, *n* = 9)	*p*-Value
Age (years)	48.00 ± 6.00	46.67 ± 9.38	0.920
Sex (% female)	5 (55.56%)	6 (66.67%)	-
Walking aid	5 (55.56%)	4 (44.44%)	-
BMI (kg/m^2^)	23.12 ± 3.49	25.59 ± 4.37	0.310
Disease duration (years)	12.33 ± 7.31	9.89 ± 6.77	0.400
Type of MS	RRMS = 55% PPMS = 45%	RRMS = 55% PPMS = 45%	-
T25FW	11.35 ± 10.84	7.38 ± 2.11	0.546
MSWS-12	39.33 ± 10.90	30.56 ± 11.33	0.077
TUG	12.08 ± 5.25	10.27 ± 2.49	0.489
EDSS total	4.94 ± 1.21	3.83 ± 1.20	0.400
EDSS pyramidal	3.33 ± 1.23	2.89 ± 0.93	0.489
MSQol-54 Physical	55.27 ± 19.70	59.41 ± 16.72	0.190
MSQol-54 Mental	44.24 ± 19.48	58.29 ± 18.60	0.161
AQL	6.22 ± 2.39	5.89 ± 1.45	0.796

AQL: Analogic Quality of Life scale; BMI = Body Mass Index; EDSS: Expanded Disability Status Score; MSQol-54: The Multiple Sclerosis Quality of life-54 questionnaire; MSWS-12: Multiple Sclerosis Walking Scale; PPMS = Primary Progressive Multiple Sclerosis; RRMS = Relapsing Remitting Multiple Sclerosis; T25FW: Timed 25-Foot Walk; TUG: Timed up and go test. Data are presented as mean ± standard deviation.

**Table 2 healthcare-12-00010-t002:** Within- and between-group comparisons of all measures at different time points.

Variable	Group	Descriptive Data				Within-Group Differences				Between-Group Differences		
		Baseline	Post	1 Week	4 Weeks	Global ^a^	Post ^b^	1 Week ^b^	4 Weeks ^b^	Post ^c^	1 Week ^c^	4 Weeks ^c^
T25FW	IG SG	11.35 ± 10.84 7.38 ± 2.11	9.81 ± 8.18 7.01 ± 1.84	11.69 ± 12.64 7.18 ± 1.83	10.83 ± 9.97 6.95 ± 1.92	*p* = 0.042 * *p* = 0.082	−1.54 ± 2.66 (*p* = 0.008) * −0.37 ± 0.13 (*p* = 0.082)	0.34 ± 1.80 (*p* = 0.515) −0.20 ± 0.10	−0.52 ± 0.87 (*p* = 0.260) −0.43 ± 0.19	2.8 ± 6.34 (*p* = 0.077)	4.51 ± 10.81 (*p* = 0.489)	3.88 ± 8.14 (*p* = 0.863)
MSWS-12	IG SG	39.33 ± 10.90 30.56 ± 11.33	- -	35.89 ± 11.26 31.11 ± 13.93	36.33 ± 9.98 32.22 ± 15.01	*p* = 0.002 * *p* = 0.962	- -	−3.44 ± 0.36 (*p* = 0.017) * 0.55 ± 0.60 (*p* = 0.962)	−3.00 ± 0.93 (*p* = 0.011) * 1.66 ± 3.68	-	4.78 ± −2.67 (*p* = 0.063)	4 ± −5.03 (*p* = 0.063)
TUG	IG SG	12.08 ± 5.25 10.27 ± 2.49	11.06 ± 5.28 9.41 ± 2.08	11.29 ± 6.91 10.03 ± 2.93	13.26 ± 10.35 9.82 ± 2.91	*p* = 0.015 * *p* = 0.091	−1.02 ± 0.03 (*p* = 0.008) * −0.86 ± 0.10 (*p* = 0.091)	−0.79 ± 1.65 (*p* = 0.110) −0.24 ± 0.44	1.18 ± 5.10 (*p* = 0.678) −0.45 ± 0.41	1.65 ± 3.21 (*p* = 0.666)	1.26 ± 3.97 (*p* = 0.113)	3.44 ± 7.44 (*p* = 0.931)
EDSS Total	IG SG	4.94 ± 1.21 3.83 ± 1.20	- -	- -	4.94 ± 1.21 3.83 ± 1.20	*p* = 1.000 *p* = 1.000	- -	- -	No change No change	-	-	1.11 ± 0.01 (*p* = 1.000)
EDSS Pyramidal	IG SG	3.33 ± 1.23 2.89 ± 0.93	- -	- -	3.44 ± 1.13 2.89 ± 0.93	*p* = 0.317 *p* = 1.000	- -	- -	0.11 ± 0.10 No change	-	-	0.55 ± 0.20 (*p* = 0.730)
MSQol-54 Physical	IG SG	55.27 ± 19.71 59.41 ± 16.72	- -	- -	61.50 ± 23.43 59.00 ± 18.67	*p* = 1.000 *p* = 1.000	- -	- -	6.23 ± 3.73 (*p* = 0.014 *) −0.41 ± 1.95 (*p* = 0.096)	-	-	6.23 ± 3.95 (*p* = 0.014) *
MSQol-54 Mental	IG SG	44.24 ± 19.48 58.29 ± 18.60	- -	- -	46.76 ± 19.19 55.82 ± 19.56	*p* = 0.096 *p* = 0.096	- -	- -	2.52 ± 0.12 (*p* = 0.096) −2.47 ± 0.96 (*p* = 0.096)	-	-	2.52 ± −0.29 (*p* = 0.297)
AQL	IG SG	6.22 ± 2.39 5.89 ± 1.45	- -	6.02 ± 1.81 6.11 ± 1.27	6.56 ± 2.12 6.22 ± 1.48	*p* = 0.001 * *p* = 0.001 *	- -	−0.18 ± −0.58 (*p* =0.441) 0.22 ± −0.18 (*p* =0.953)	0.54 ± 0.18 (*p* =0.008 *) 0.33 ± 0.03 (*p* =0.008 *)	- -	0.09 ± −0.54 (*p* = 0.863)	−0.34 ± −0.64 (*p* = 1.000)

AQL: Analogic Quality of Life scale; EDSS: Expanded Disability Status Score; IG: Intervention Group; MSQol-54: The Multiple Sclerosis Quality of life-54 questionnaire; MSWS-12: Multiple Sclerosis Walking Scale; SG: Sham Group; T25FW: Timed 25-Foot Walk; TUG: Timed up and go test. Data are represented as mean ± standard deviation. * Statistically significant results (*p* < 0.05); ^a^ Non-parametric Friedman test (changes over time in each group); ^b^ Non-parametric Wilcoxon test (carried out on those variables in which the overall result of the Friedman test was significant); ^c^ Non-parametric Mann–Whitney U test (comparisons between groups).

## Data Availability

The datasets used and/or analyzed during the current study available from the first author on reasonable request by email to albert.jaor@gmail.com.

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
