# Peer review of "The Effectiveness of a Single Dry Needling Session on Gait and Quality of Life in Multiple Sclerosis: A Double-Blind Randomized Sham-Controlled Pilot Trial"

_healthcare, 2023, doi:10.3390/healthcare12010010_

Round 1

Reviewer 1 Report

Comments and Suggestions for Authors

Dear editor, first of all, thank you for the invitation you sent me to review this study entitled “Effectiveness of a single dry needling session on gait and quality of life in multiple sclerosis: a double-blind randomized  sham-controlled pilot trial”.

This is a prospective, randomized controlled study. Authors aimed to  evaluate the effect of a single session of dry in the gait performance of patients with MS. I congratulate researchers for their work.

Overall, the article is written in a clear, fluent and understandable language.

I have some suggestion to improve the article.

Abstract:

I suggest to authors revise conclusion part of abstract according to their study results, because they stated that “However, when 245 results were compared with sham DN, no differences were found. “ I think this is important.

Main text: 

I suggest to the authors to give more information about physiotherapy program which patients received. 

Although each patient is different, information about the duration of the session, the frequency of the session and the treatment content in general can be provided. 

Was there a patient using a walking aid?

Was the gastrocnemius muscle affected in each patient? Some patients may have had impaired balance, others may have had impaired gait function because their optic nerve was affected. In this sense, I think there are many parameters that will affect the results of the study.

I think the most important point in this study, and the one that was missed, is that the authors failed to discuss the impact of dry needling. When we do dry needling, what effect does it have and how does it affect gait parameters?

What is the purpose of dry needling here? What effect did it have on the tissues? Maybe it would be better to start with a hypothesis? Please add them to the discussion.

I suggest you add a photo of the application if you have it available.

Author Response

Abstract:

I suggest to authors revise conclusion part of abstract according to their study results, because they stated that “However, when 245 results were compared with sham DN, no differences were found. “ I think this is important.

Response: The text has been added in abstract

Main text: 

I suggest to the authors to give more information about physiotherapy program which patients received. 

Response: Patients only received one session of dry needling; they did not receive any other physical therapy intervention. We have detailed this in the text.

Although each patient is different, information about the duration of the session, the frequency of the session and the treatment content in general can be provided. 

Response: Participants received a single session of active or sham dry needling for 30 seconds. All sessions lasted the same for all participants. This was described in the methodology section and we followed the same protocol without doing any exceptions.

Was there a patient using a walking aid?

Response: The percentage of patients who used some type of walking aid has been added to Table 1

Was the gastrocnemius muscle affected in each patient? Some patients may have had impaired balance, others may have had impaired gait function because their optic nerve was affected. In this sense, I think there are many parameters that will affect the results of the study.

Response: In the inclusion criteria, it was established to have a pyramidal value of the EDSS scale greater than 2. This criterion involves that muscles will be affected to a greater or lesser extent in the lower limbs, including the gastrocnemius medialis. We have detailed this criterion better in the methodology.

I think the most important point in this study, and the one that was missed, is that the authors failed to discuss the impact of dry needling. When we do dry needling, what effect does it have and how does it affect gait parameters?

What is the purpose of dry needling here? What effect did it have on the tissues? Maybe it would be better to start with a hypothesis? Please add them to the discussion.

Response: We have a added a paragraph to support the effects of DN in spastic muscles and gait speed in other neurological conditions and mentioned the neuromodulation effects that DN may have to support the possibility to have changes not only in quantitative but also qualitative parameters of gait.

I suggest you add a photo of the application if you have it available.

Response: The photo has been added as Fig. 1

Reviewer 2 Report

Comments and Suggestions for Authors

1. information about young age twice (43,47)

2. Multiple sclerosis affects the brain, brainstem, spinal cord and optic nerves, potentially causing weakness, spasticity and fatigue, as well as changes in sensation, coordination, vision and cognitive function. It is therefore not surprising that imbalance, gait disturbances and falls are common in people with multiple sclerosis (pwMS) (56) - a correction proposal

once again the information accidental falls (57,69)

3. PwMS walk slower than age-matched controls, with a lower cadence and increased stride-to-stride variability (70) - whether the control group was healthy or with MS, because comparing a healthy group to a sick group will always indicate a disorder, it is rather predictable .

4. Therefore, the improvement of walking ability is a primary goal for rehabilitation treatment in pwMS (71) - the patient has the ability to walk, possibly it can be improved quantitatively

5. The main focus of rehabilitation treatments in pwMS patients is improving weakness balance and gait (72) - weakness or balance disorders

6. It is worth describing the exact place of intervention for the research and control groups (maybe a photo)

Author Response

  1. information about young age twice (43,47)

Response: It appears twice because one is the total population age and the other divided into groups

  1. Multiple sclerosis affects the brain, brainstem, spinal cord and optic nerves, potentially causing weakness, spasticity and fatigue, as well as changes in sensation, coordination, vision and cognitive function. It is therefore not surprising that imbalance, gait disturbances and falls are common in people with multiple sclerosis (pwMS) (56) - a correction proposal

once again the information accidental falls (57,69)

Response: The paragraph has been rewritten to avoid repetitions and also following the editor´s suggestions.

  1. PwMS walk slower than age-matched controls, with a lower cadence and increased stride-to-stride variability (70) - whether the control group was healthy or with MS, because comparing a healthy group to a sick group will always indicate a disorder, it is rather predictable .

Response: we have rewritten the whole paragraph.

  1. Therefore, the improvement of walking ability is a primary goal for rehabilitation treatment in pwMS (71) - the patient has the ability to walk, possibly it can be improved quantitatively

Response: we have highlighted some quantitative parameters of gait (cadence and speed as well as stride to stride variability) as the most relevant

  1. The main focus of rehabilitation treatments in pwMS patients is improving weakness balance and gait (72) - weakness or balance disorders

Response: we didn´t understand what the reviewer asked here. The sentence says that “The main focus of rehabilitation treatments in pwMS patients is improving weakness, balance and gait” so it can be understood that there are some disorders in balance and gait, which have been previously mentioned in the previous paragraph.

  1. It is worth describing the exact place of intervention for the research and control groups (maybe a photo)

Response: We have added a photo (Fig. 1)

Round 2

Reviewer 1 Report

Comments and Suggestions for Authors

The authors have made the necessary revisions.

Congratulations for their work.